# How honey bees make fast and accurate decisions

**HaDi MaBouDi[1,2]*[†], James AR Marshall[1,2][†], Neville Dearden[1], Andrew B Barron[1,3]**

[1]Department of Computer Science, University of Sheffield, Sheffield, United Kingdom; [2]Sheffield Neuroscience Institute, University of Sheffield, Sheffield, United Kingdom; [3]School of Natural Sciences, Macquarie University, North Ryde, Australia

**Abstract** Honey bee ecology demands they make both rapid and accurate assessments of which flowers are most likely to offer them nectar or pollen. To understand the mechanisms of honey bee decision-making, we examined their speed and accuracy of both flower acceptance and rejection decisions. We used a controlled flight arena that varied both the likelihood of a stimulus offering reward and punishment and the quality of evidence for stimuli. We found that the sophistication of honey bee decision-making rivalled that reported for primates. Their decisions were sensitive to both the quality and reliability of evidence. Acceptance responses had higher accuracy than rejection responses and were more sensitive to changes in available evidence and reward likelihood. Fast acceptances were more likely to be correct than slower acceptances; a phenomenon also seen in primates and indicative that the evidence threshold for a decision changes dynamically with sampling time. To investigate the minimally sufficient circuitry required for these decision-making capacities, we developed a novel model of decision-making. Our model can be mapped to known pathways in the insect brain and is neurobiologically plausible. Our model proposes a system for robust autonomous decision-making with potential application in robotics.

**\*For correspondence:**
maboudi@gmail.com

**Present address:** [†]Opteran Technologies, Sheffield Innovation Centre, Sheffield, United Kingdom

**Competing interest:** The authors declare that no competing interests exist.

## Editor's evaluation

This valuable study elucidates the honeybee's behavioral strategy to associate sensory cues with rewards of different values. Based on solid experimental evidence the study demonstrates how sensory evidence and reward likelihood quantitatively affect the decision-making process and the bees' response time. The behavioral paradigm and the proposed model could provide interesting insights for scientists studying decision-making in higher animal species.

## Introduction

Decision-making is at the core of cognition. A decision can be considered as the result of an evaluation of possible outcomes (*Mobbs et al., 2018*; *Stevens, 2011*), and animal lives are full of decisions. What we might consider to be a simple choice, for example choosing the best option from two alternatives, is rarely simple in an ecological setting (*Mobbs et al., 2018*). Consider the decisions a foraging bee makes. A bee, moment by moment, must decide whether a flower should be explored for pollen and nectar or whether it is not worth landing on. We could suppose that decision to be influenced by what the bee can sense about the flower, her past experiences with that flower type, the context (is a predator nearby?), the state of the bee (does she already carry a full load of nectar and pollen?) and the state of her colony (what does the colony need?) (*Chittka, 2022*; *Conradt and Roper, 2005*; *Stephens, 2008*). Even this simple decision is a whole-brain activity involving sensory systems, memory systems, motor systems, and the bee's subjective state. Here, we studied honey bee foraging decisions in controlled conditions to establish their decision-making capacities. We then

**eLife digest** In the natural world, decision-making processes are often intricate and challenging. Animals frequently encounter situations where they have limited information on which to rely to guide them, yet even simple choices can have far-reaching impact on survival.

Each time a bee sets out to collect nectar, for example, it must use tiny variations in colour or odour to decide which flower it should land on and explore. Each 'mistake' is costly, wasting energy and exposing the insect to potential dangers. To learn how to refine their choices through trial-and-error, bees only have at their disposal a brain the size of a sesame seed, which contains fewer than a million neurons. And yet, they excel at this task, being both quick and accurate. The underlying mechanisms which drive these remarkable decision-making capabilities remain unclear.

In response, MaBouDi et al. aimed to explore which strategies honeybees adopt to forage so effectively, and the neural systems that may underlie them. To do so, they released the insects in a 'field' containing artificial flowers in five different colours. The bees were trained to link each colour with a certain likelihood of receiving either a sugary liquid (reward) or bitter quinine (punishment); they were then tested on this knowledge.

Next, MaBouDi et al. recorded how the bees would navigate a 'reduced evidence' test, where the colour of the flowers were ambiguous and consisted in various blends of the originally rewarded or punished colours; and a 'reduced reward likelihood' test, where the sweet recompense was offered less often than before.

Response times and accuracy rates revealed a complex pattern of decision-making processes. How quickly the insects made a choice, and the types of mistakes they made (such as deciding to explore a non-rewarded flower, or to ignore a rewarded one) were dependent on both the quality of the evidence and the certainty of the reward. Such sophistication and subtlety in decision-making is comparable to that of primates.

Next, MaBouDi et al. developed a computational model which could faithfully replicate the pattern of decisions exhibited by the bees, while also being plausible biologically. This approach offered insights into how a small brain could execute such complex choices 'on the fly', and the type of neural circuits that would be required. Going forward, this knowledge could be harnessed to design more efficient decision-making algorithms for artificial systems, and in particular for autonomous robotics.

developed a simple model with the same capacities for decision-making as a bee to assist in hypothesising the necessary neural mechanisms supporting bees' foraging decisions.

Abstract theories and models of decision-making are well-developed, and these provide frameworks for evaluating animals' decision-making capacity (*Gold and Shadlen, 2007*; *Mobbs et al., 2018*; *O'Connell et al., 2018*). Here, we apply signal detection theory to understand how bees make a decision (*Green and Swets, 1966*; *Green and Swets, 1966*; *Sumner and Sumner, 2020*; *Wickens, 2001*). Signal detection theory helps us think formally about the processes of signal discrimination, which is essential for making decisions (*Wickens, 2001*). It provides an abstract model and simple logic for how animals should respond given the signal they have received and their prior knowledge. Typically signal detection theory assumes that an individual must choose between two possible actions (acceptance or rejection) after detecting a signal. In such a scenario, there are four possible outcomes, which include two correct actions. These are: 1, correct acceptance when the subject accepts the correct stimulus ('hit'), 2, correct rejection when the subject rejects the incorrect stimulus (correct rejection), 3, incorrect acceptance when the subject wrongly accepts the incorrect stimulus ('false positive', Type I error), 4, incorrect rejection when the subject rejects the correct stimulus ('false negative', Type II error). The optimal decision is calculated by considering the expected payoffs of all four outcomes together. Both errors are integral parts of the decision-making process. In an ecological context, both errors typically differ in costs to an animal (*Sumner and Sumner, 2020*). For example, wrongly rejecting a food item might see an animal missing a meal, but wrongly accepting a food item could see an animal ingesting poison. Signal detection theory emphasises that both acceptance and rejection choices have to be assessed if decision-making is to be understood, but typically in studies of animal behaviour rejection behaviour is ignored (*Ings and Chittka, 2008*; *Sumner and Sumner, 2020*; *Trimmer et al., 2017*).

Decision-making processes are most often modelled with sequential sampling models, of which there are many variations (*O'Connell and Hofmann, 2012*; *O'Connell et al., 2018*). Sequential sampling models are built on the biologically realistic assumptions that sensory information on available options is noisy, but evidence for different options accumulates over time through sequential sampling (*Gold and Shadlen, 2007*). A decision is made when the cumulant reaches a threshold. Variations in sequential sampling models differ in the nature of the threshold for the decision. For example, in the race model (*Vickers, 1970*) a decision is made when evidence for one alternative reaches an upper threshold. Leaky competing accumulator (LCA) models set the evidence for different options in competition such that as evidence for one option accumulates it inhibits evidence for the alternative and a decision is made when the difference in evidence for the two alternatives reaches a threshold (*Barron et al., 2015*; *Bogacz et al., 2006*). Sequential sampling models have proved very influential in neuroscience, psychology, and computer science. While they are highly abstract, they capture many features of biological decision-making, particularly a speed/accuracy trade-off (*Barron et al., 2015*; *Bogacz et al., 2006*; *Gold and Shadlen, 2007*; *Pirrone et al., 2014*).

Investigation of the neural mechanisms of choice in primates has revealed interacting neural systems for the evaluation of different options and the selection of a choice that involve the frontal cortex, the basal ganglia, and the frontal and parietal cortices (*Barron et al., 2015*; *Gurney et al., 2001*; *Seed et al., 2011*; *Shadlen and Kiani, 2013*; *Wang, 2012*). This is a system of extreme complexity, involving billions of neurons. Most animal brains are orders of magnitude smaller than this. How might smaller brains make effective decisions? To this end, we explored honey bee foraging decisions. We measured bees' acceptance and rejection of different options under controlled conditions that manipulated the quality of available evidence and the probability of a rewarding outcome. To understand the properties of bee decision-making, we explored our data with signal detection theory and also examined how accuracy varied with decision speed. Having identified the key properties of bee decision-making we then constructed the simplest sequential sampling model capable of the same decision-making capacities as the bee. Finally, we related this abstract model to the known systems of the bee brain to propose a hypothetical brain mechanism for autonomous decision-making in insects.

## Results

We individually trained 20 honey bees (*Apis mellifera*) on a colour discrimination task in which they learned to associate five distinct colours each with their visit history of reward and punishment. Over 18 training trials, each colour offered bees a different likelihood of reward and punishment (*Figure 1A*, *Figure 1—source data 1*). The five colours offered the reward in 100%, 66%, 50%, 33%, and 0% of training trials (*Figure 1B*) and were otherwise punished. The colour rewarded in 100% of training trials was never punished while the colour rewarded in 0% of training trials was always punished. Each trial offered bees just one pair of colours with one colour in the pair rewarded more often than the other during training (See Materials and methods, *Figure 1—source data 1*, *Figure 2—figure supplement 1A,B*, *Table 1*). Following training, bees were given three tests. In the *easy discrimination test*, each honey bee was tested with the two colours rewarded at 100% and 0% in training. In the *reduced evidence test,* bees were tested with two novel colours that were different blends of blue and green (the 100% and 0% rewarded colours) to determine how behaviour changed when the available evidence was degraded. One blend was closer to blue and one closer to green. In the *reduced reward likelihood test* bees were presented with the 66% and 33% rewarded colours to assess how bees' behaviour changed when the likelihood of reward offered by a choice was less than 100%. In the easy discrimination and reduced evidence tests, correct choices were considered as acceptance of the more rewarded colour, and rejection of the less rewarded colour. Bee's acceptance and rejection responses were analysed from videos recorded during the training and tests (*Figure 1D*, see Materials and methods section). We employed the Matthew Correlation Coefficient (MCC) (*MaBouDi et al., 2020a*) to measure the performance of the bees in each test. This considered all types of responses (i.e. hit, correct rejection, false positive, and false negative) to calculate decision accuracy such that a positive correlation (with a maximum value of +1) indicates perfect performance accuracy while a value of zero indicates chance-level performance. Values between 0 and +1 demonstrate varying degrees of decision accuracy (see Materials and methods section).

In our free-flight choice assay bees learned to prefer the 100% rewarded colour from the 0% rewarded colour (*Figure 2A*; Wilcoxon signed rank test: z=3.62, n=20, p=2.93e-4; see *Figure 2—figure*

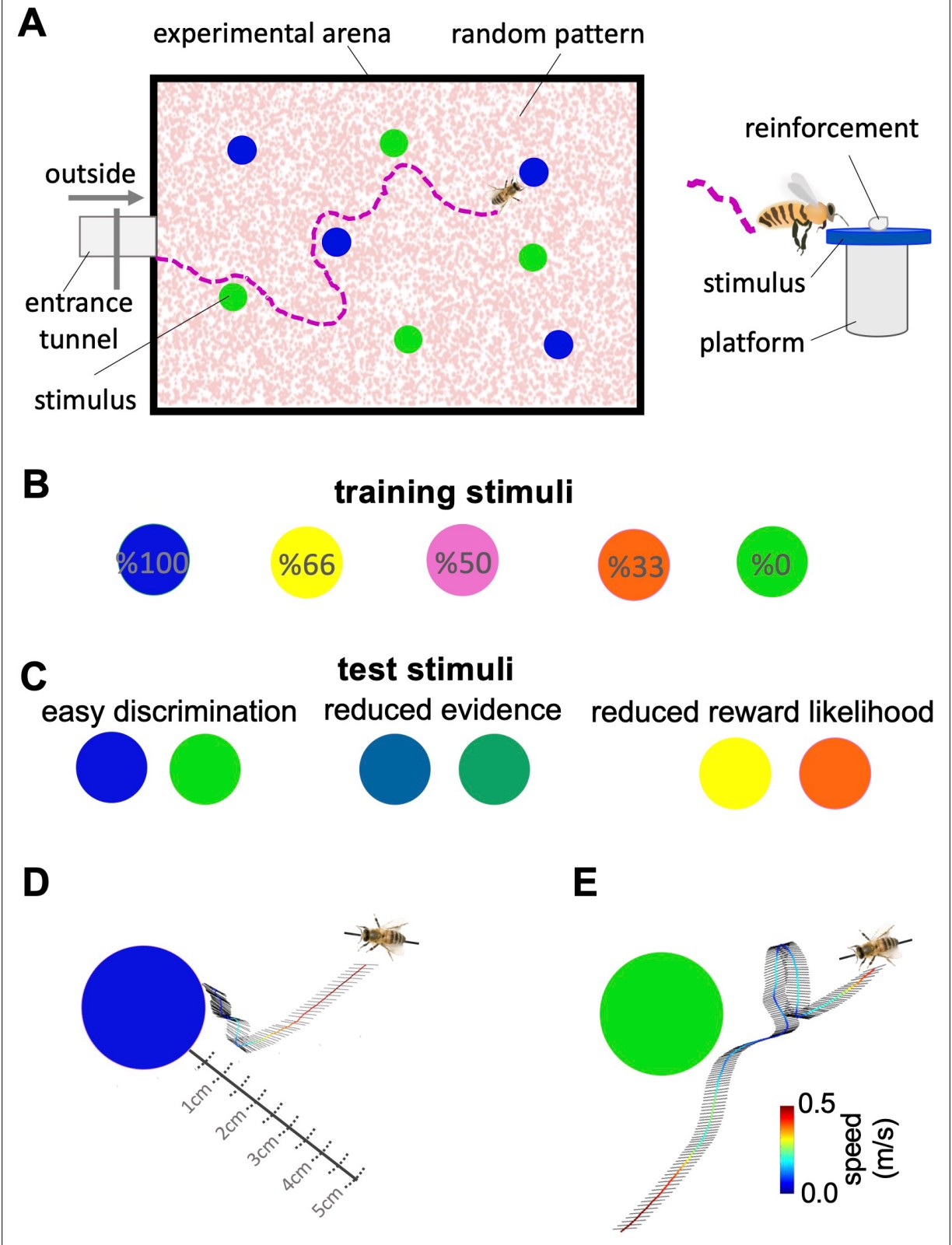

**Figure 1.** Bees' behaviour in a colour discrimination task. (**A & B**) Each bee was given 18 training trials in which she could choose between two different colours: one rewarded and the other punished. The bee was free to select each colour and return to the hive when satiated marking the end of a trial. Stimuli positions in the arena were changed in each trial in a pseudo-random manner. Stimuli were 2 cm diameter-coloured disks on a small platform (5 cm tall). On the top of each colour was placed either 10 µl reward (50% sucrose) or punishment (quinine) in training, or distilled water in tests. Two

*Figure 1 continued on next page*

*Figure 1 continued*

different colours, four disks of each colour, were presented in each training trial and test. Five different colours were used in the training. The colours differed in the proportion of training bouts in which they offered reward and punishment (rewarded at 100, 66, 50, 33, and 0% of training trials). Two groups of bees were trained with different likelihoods of reward and punishment from each colour (see Materials and methods section and *Figure 1— source data 1*). (C) Following training, the bee was given three unreinforced tests where the positive or negative reinforcements were replaced with distilled water. Bees' responses were analysed from video recordings of the first 120 s in the flight arena. In the easy colour discrimination test, bees were presented with three pairs of the 100% and 0% rewarded colours (blue and green). In the reduced reward likelihood test, bees were examined with 66% and 33% rewarded colours (yellow and orange). In the reduced evidence test. bees were given two colours intermediate between green and blue (D & E) Examples of flight paths showing the inspection activity of a bee during the easy discrimination test in accepting blue (D) and rejecting green (E). Each black line on the flight path corresponds to the bee's body orientation in a single video frame with 4ms intervals between frames. Line colour: flight speed 0.0–0.5 m/s (See *Video 1*).

The online version of this article includes the following source data for figure 1:

**Source data 1.** Bees' choices during the training trials.

supplement 1C for power analysis). Bees' performance in the reduced evidence test was lower but was still higher than chance (*Figure 2A*; Wilcoxon signed rank test: z=2.10, n=18, p=0.03). In the reduced reward likelihood test, bees selected the 66% reward colour more frequently than chance (*Figure 2—figure supplement 2*).

Bees spent longer in flight before their first landing in the tests than in the first training trial (*Figure 2B*; Kruskal-Wallis test, chi-sq=13, df = 7, p=4.60e-3). This shows that during training bees developed a behaviour of assessing the available stimuli in the arena for longer before landing. There was a significant negative correlation between bees' performance in the easy discrimination test and their time to first landing (assessed by the MCC: Spearman correlation, rho = –0.55, n=20, p=0.02). Poor performance in the test was associated with a longer time before a first choice (*Figure 2C*).

**Table 1.** Two different sequences of training trials were used.
10 bees were trained with the protocol P1 and 10 with the protocol P2.

| | Protocol P1 | | Protocol P2 |
|---|---|---|---|
| #trials | colours at each trial | #trials | colours at each trial |
| 1 | S100% vs S66% | 1 | S50% vs S0% |
| 2 | S50% vs S0% | 2 | S100% vs S66% |
| 3 | S100% vs S33% | 3 | S100% vs S33% |
| 4 | S66% vs S0% | 4 | S66% vs S0% |
| 5 | S50% vs S33% | 5 | S100% vs S50% |
| 6 | S100% vs S50% | 6 | S50% vs S33% |
| 7 | S33% vs S0% | 7 | S100% vs S0% |
| 8 | S66% vs S50% | 8 | S33% vs S0% |
| 9 | S100% vs S0% | 9 | S66% vs S50% |
| 10 | S100% vs S66% | 10 | S100% vs S0% |
| 11 | S50% vs S0% | 11 | S50% vs S33% |
| 12 | S100% vs S33% | 12 | S66% vs S50% |
| 13 | S66% vs S0% | 13 | S33% vs S0% |
| 14 | S50% vs S33% | 14 | S100% vs S50% |
| 15 | S100% vs S50% | 15 | S66% vs S0% |
| 16 | S33% vs S0% | 16 | S100% vs S33% |
| 17 | S66% vs S50% | 17 | S100% vs S66% |
| 18 | S100% vs S0% | 18 | S50% vs S0% |

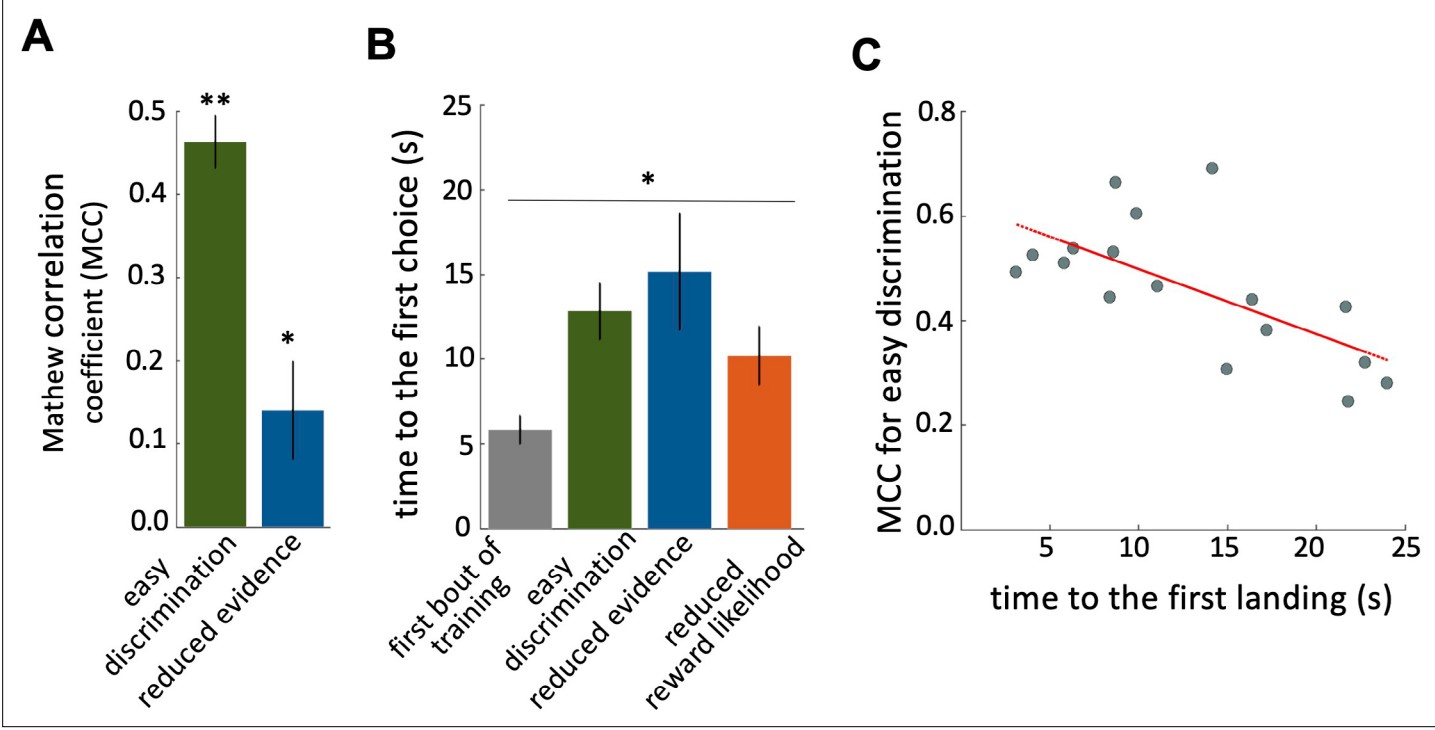

**Figure 2.** Characteristics of bee decision making. (**A**) Matthew correlation coefficients (MCC) (mean ± SEM) for the easy discrimination and reduced evidence tests. In the both easy discrimination and reduced evidence tests, this correlation is computed with respect to choosing the high-rewarded colours for each bee. A positive correlation (max at +1) indicates perfect correct performance while zero indicates chance level performance. Correlation coefficients were significantly greater than zero for both tests. (**B**) Average time to the first choices for three tests and the first training trial. Bees naive to the stimuli made their first choice faster than bees trained on the stimuli (p=1.55e-3). (**C**) Scatter plot showing a negative correlation between the MCC and the time to first acceptance in the easy discrimination test. A rapid first choice correlated with higher performance. Values for each individual bee are shown by small circles. n=20, **p<0.005 and *p<0.05.

The online version of this article includes the following figure supplement(s) for figure 2:

**Figure supplement 1.** Bees' performance during the training.

**Figure supplement 2.** Bees' performance on the reduced reward likelihood test.

## Investigation of bee decision-making using classical signal detection theory

Signal detection theory provides a framework for understanding and predicting how animals make decisions under uncertainty by modelling the relationship between the sensory information they received and their ability to accurately discriminate between stimuli. Hence, the probability of a stimulus being correctly identified is assumed to be a function of the sensory information received. If we have two different stimuli (in our case the high and low rewarded colours) we can model how the probability of identifying them changes as perceived colour information is sampled from two overlapping normal distributions (*Figure 3A*). For each colour, it could be identified correctly or incorrectly. For a trained bee we would recognise this as four types of behavioural response. For the highly rewarded colour, these would be correct acceptance or incorrect rejection. For the low rewarded colour these would be correct rejection or incorrect acceptance (*Figure 3A*). Discriminability (d') is the difference in the sensory information between the maximal probability of the two different stimuli (*Figure 3A*). From our data, we could calculate discriminability following *Sumner and Sumner, 2020* by modelling total accept and reject responses as cumulative distribution functions and considering the hit rate (correct acceptance / total acceptance) and the false positive rate (incorrect rejections/ total rejections; *Equation 2*, Materials and methods).

When considering contrasting responses to two different stimuli using signal detection theory we can identify a threshold sensory signal at which behaviour should shift from acceptance to rejection. This is the decision criterion (*d.c.*, *Figure 3A*). From our experimental data we can estimate the

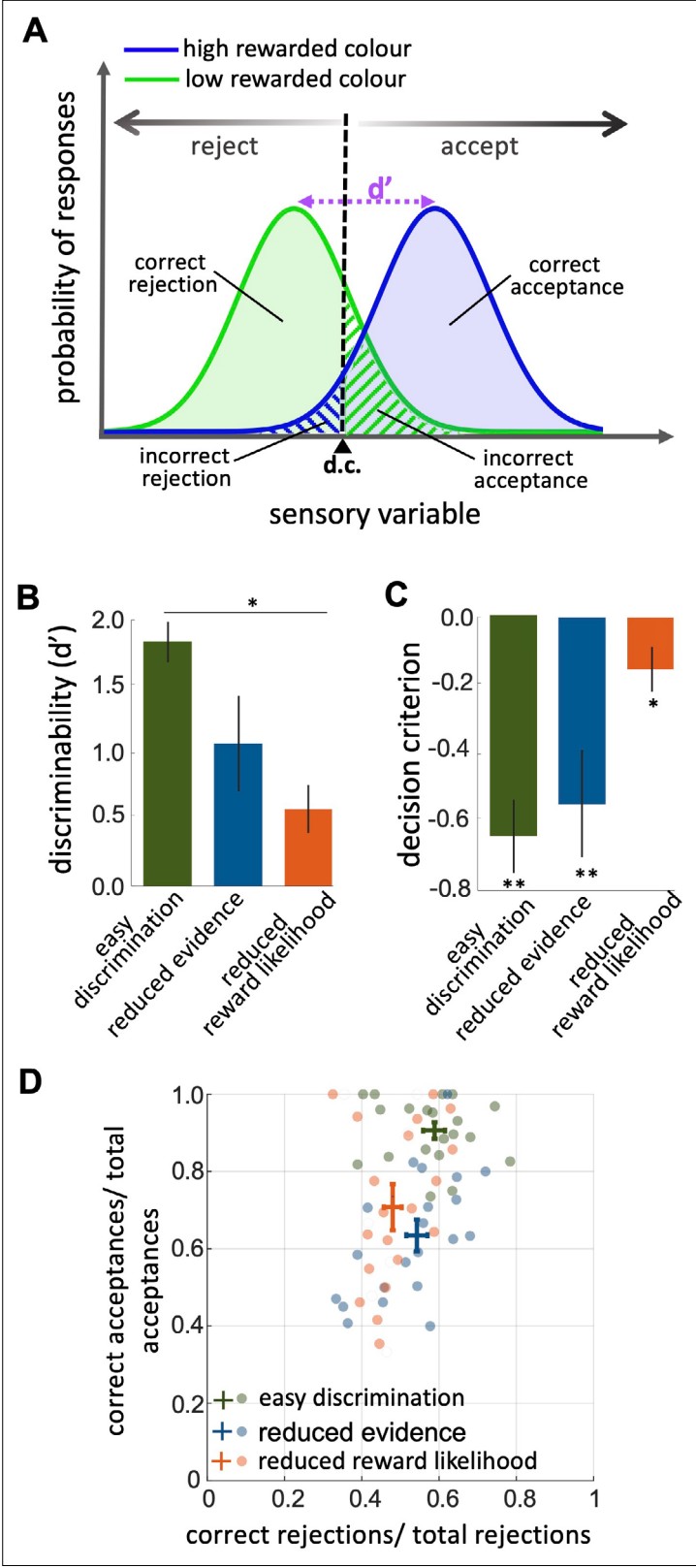

**Figure 3.** An investigation by classical signal detection theory. (**A**) Probability of responding to the high (blue) and low (green) rewarded stimuli at different levels of sensory input. For a trained bee we recognise a threshold (decision criterion, d.c.) at which their behaviour shifts from rejection to acceptance. As a result, we have four types of behavioural responses. d' is the discriminability of the two stimuli. (**B**) Discriminability was greatest in the

*Figure 3 continued on next page*

*Figure 3 continued*

easy discrimination task and was reduced in both reduced evidence and reduced reward likelihood tests. (**C**) The decision criterion was negative for the easy discrimination and reduced evidence tests indicating fewer incorrect acceptances than incorrect rejections in these tests. The decision criterion was closer to zero in the reduced reward likelihood test indicating similar accuracy of acceptance and rejection in this test. (**D**) Plotting the ratio of correct to incorrect acceptances and rejections (crosses show the mean and SEM) for the three tests show that generally, bees were more accurate in acceptance than rejection responses. Acceptance accuracy fell in the reduced evidence and reduced reward likelihood tests. n = 20, **p<0.005 and *p<0.05.

relative location of the *d.c.* by considering both the hit rate and the false positive rate (**Wickens, 2001**, **Equation 3** in the Materials and methods section). A value of 0 for the *d.c.* indicates that there were as many incorrect rejections as there were incorrect acceptances, or that the acceptance and rejection responses were equally accurate. A negative value for the decision criterion (*d.c.*) would move the decision criterion to the left in **Figure 3A**. This would result in more correct acceptances (i.e. the area under the probability of responding to the high rewarded stimuli (blue) is increased) but fewer correct rejections (i.e. the area under the probability of responding to the low rewarded stimuli [green] is decreased). It would indicate acceptance responses are more precise than rejections.

The reduced evidence test significantly decreased the discriminability of more and less rewarded stimuli (**Figure 3B**; Wilcoxon rank sum test: z=1.81, n=20, p=0.03). Discriminability was also reduced in the reduced evidence test in which the two stimuli were closer in their likelihood of being rewarded (**Figure 3B**; Wilcoxon rank sum test: z=3.94, n=20, p=8.01e-5). This shows that for bees' discriminability is influenced by both available evidence and reward likelihood.

When the likelihood of reward for the two stimuli was more similar the decision criterion was closer to zero (**Figure 3C**; Wilcoxon signed rank test: z=−2.21, n=20, p=8.4e-3) indicating that the accuracy of acceptance and rejection were more similar when the reward outcomes for the two stimuli were more similar. Otherwise, in both the easy discrimination and reduced evidence tests (in which one stimulus was always rewarded and one punished) acceptance was more accurate than rejection (**Figure 3C**; Wilcoxon signed rank test: z=−3.62, n=20, p=2.93e-4 for easy discrimination test, z=−2.91, n=18, p=3.5e-3 for reduced evidence test). Finally, the comparison of the ratio of correct and incorrect acceptance and rejection in the three tests (**Figure 3D**) revealed that the acceptance accuracy in both reduced evidence and reduced likelihood tests decreased compared to the easy discrimination test, indicating that acceptance accuracy was sensitive to both evidence and reward likelihood. Overall rejection accuracy was lower than acceptance accuracy. Rejection accuracy was lowest in the reduced reward likelihood test than in the reduced evidence test, indicating the rejection accuracy was more influenced by reward likelihood than available evidence (**Figure 3D**). This indicates that the evidence thresholds for accept and reject decisions were distinct, as discussed further in the Discussion section.

## How quality of evidence and reward likelihood influence decision accuracy and decision speed

In the easy discrimination test, there were more rejections than acceptances (**Figure 4B**; Wilcoxon signed rank test: z=−3.62, n=20, p=2.9e-4) and bees' accuracy (the difference between the number of correct and incorrect choices) of acceptance was higher than rejection (**Figure 4B**; Wilcoxon signed rank test: z=3.42, n=20, p=6.1e-4). Also, bees' accuracy of acceptance in the easy discrimination test was higher than bees' responses in the reduced evidence test (**Figure 4B and C**; Wilcoxon signed rank test: z=3.77, n=18, p=1.57e-4). While the number of correct rejections is higher than the number of incorrect rejection responses in the easy discrimination test (**Figure 4B**; Wilcoxon signed rank test: z=1.94, n=20, p=0.43), in the reduced evidence test there was no difference in the number of correct and incorrect rejection responses (**Figure 4C**; Wilcoxon signed rank test: z=−0.66, n=20, p=0.50). Hence, we propose that acceptance responses are more accurate than rejection responses, but reducing the available evidence reduced the capacity of bees to distinguish the correct and incorrect options.

Classical signal detection theory does not consider how signals might be influenced by sampling time, but in our data, we noticed bees differed in the time they spent inspecting stimuli. To explore this, we analysed how bees' response times influenced their choices.

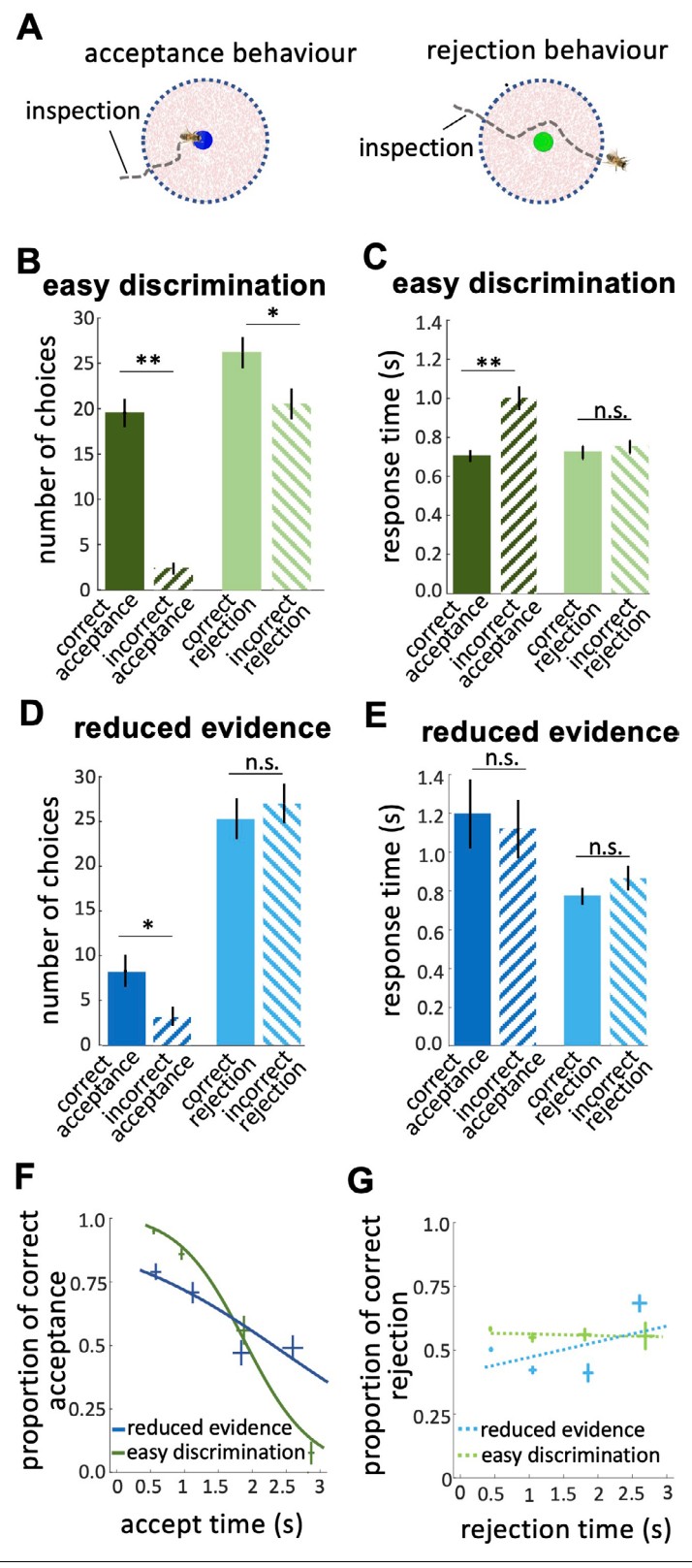

**Figure 4.** Response times of bee decisions. (**A**) Bees inspected the coloured stimuli prior to accepting or rejecting a colour. (**B**) The number of rejections was higher than the number of acceptances in the easy discrimination test. The difference between the correct and incorrect acceptances was larger than the difference between correct and incorrect rejections. (**C**) In the easy discrimination test bees accepted correct colours faster than incorrect

*Figure 4 continued on next page*

*Figure 4 continued*

colours, but there was no difference in the response time for correct and incorrect rejections. (**D**) In the reduced evidence test there were still more correct acceptances than incorrect acceptances, but the number of correct acceptances decreased. (**E**) Acceptance times for the correct colour were increased in the reduced evidence test. Bees took longer to accept stimuli with reduced evidence comparing to rejection responses, for both correct or incorrect choices. (**F**) Conditional Accuracy Function (CAF) plot for acceptance responses in the reduced evidence and easy discrimination tests. Lines show the best fit of piece-wise logistic regressions to the bee's response time. Acceptance accuracy declined with increasing response time. The vertical and horizontal lines at each cross indicate the standard deviation of the proportion of correct acceptance and accept time, respectively. (**G**) CAF curve for rejections in both easy discrimination and reduced evidence tests. The accuracy of rejection did not change significantly with response time. The vertical and horizontal lines at each cross indicate the standard deviation of the proportion of correct rejection and rejection time, respectively. n=20, **p<0.005, *p<0.05 and n.s., p>0.05.

Prior to bees accepting or rejecting stimuli, we noticed the bees hovered close to and facing the stimulus (*Figure 4A*). We hypothesise bees were sampling information about the stimulus. In the easy discrimination test bees accepted the correct colour faster than the incorrect one (*Figure 4C*; Wilcoxon signed rank test: z=−2.62, n=20, p=8.8e-3), but rejection times did not differ for correct and incorrect colours (*Figure 4C*; Wilcoxon signed rank test: z=−0.40, n=20, p=0.68). This shows the acceptance response was more accurate than the rejection, indicating a higher level of discrimination. In the reduced evidence test there was little difference between correct and incorrect response times (*Figure 4E*; Wilcoxon signed rank test: z=−0.25, n=20, p=0.79 for acceptances; z=−1.28, n=18, p=0.19 for rejections), and longer acceptance times overall (*Figure 4E*; Wilcoxon signed rank test: z=1.98, n=18, p=0.046), suggesting bees struggled to distinguish the correct and incorrect options in the reduced evidence test.

We calculated the Conditional Accuracy Functions (CAF) for acceptance and rejection responses, which is the subject's accuracy as a function of the decision time (*Figure 4F & G*; *Murphy et al., 2016*). For each bee, we assessed the response time for all acceptance responses (both correct and incorrect) in the reduced evidence and easy discrimination tests. Response times were divided into 0.5 s bins and, for each bin, we calculated the proportion of correct acceptances as the number of correct acceptances / total acceptances in that response time bin. The negative slope of the CAF curves for acceptance indicates that bees made correct acceptances faster than incorrect acceptances (*Figure 4F*; Spearman correlation, rho = −0.43, n=20, p=3.0e-3). However, the CAF for the reduced evidence test was lower than the CAF for the easy discrimination test for almost the entire range of the response time (*Figure 4F*; Spearman correlation, rho = −0.25, n=18, p=6.5e-2). The gradient of the CAF curve was decreased by reducing the available evidence. This shows that decisions based on reduced evidence are slower and less accurate, and accuracy varied less with decision time. The CAF for the rejection response showed that rejection time did not vary with accuracy (*Figure 4G*; Spearman correlation, rho = 0.07, n=20, p=0.87 for easy discrimination test; rho = 0.02, n=18, p=0.81 for reduced evidence test). Collectively our analyses show that acceptance behaviour is very accurate and therefore very sensitive to available evidence, whereas rejection behaviour is less accurate, and hence is less sensitive to changes in evidence (See Discussion section).

## Bees' choice strategy is sensitive to the history of reward

In the reduced reward likelihood test bees were more likely to reject than accept stimuli (*Figure 5A*; Wilcoxon signed rank test: z=−3.46, n=20, p=5.35e-4). In the reduced reward likelihood test bees had experienced both stimuli as rewarded and punished (33% and 66% punished) during training. We observed acceptance and rejection responses to both stimuli, most likely because bees were displaying the strategy of matching their choices to the probability each stimulus was rewarded in training (*MaBouDi et al., 2020b*). In the reduced reward likelihood test, there was no difference in times to accept and reject (*Figure 5B*; Wilcoxon signed rank test: z=−0.51, n=20, p=0.60 for acceptances; z=−1.15, n=20, p=0.24). Comparing the acceptance time of the easy discrimination, reduced evidence and reduced likelihood reward tests showed that fast acceptance is associated with more reliable evidence and certainty of outcome, and slower acceptance times are associated with less reliable evidence or less certainty of reward (comparing *Figures 4C and 5B*). No negative slope of CAF curves was observed for either acceptance or rejection behaviour in the reduced likelihood

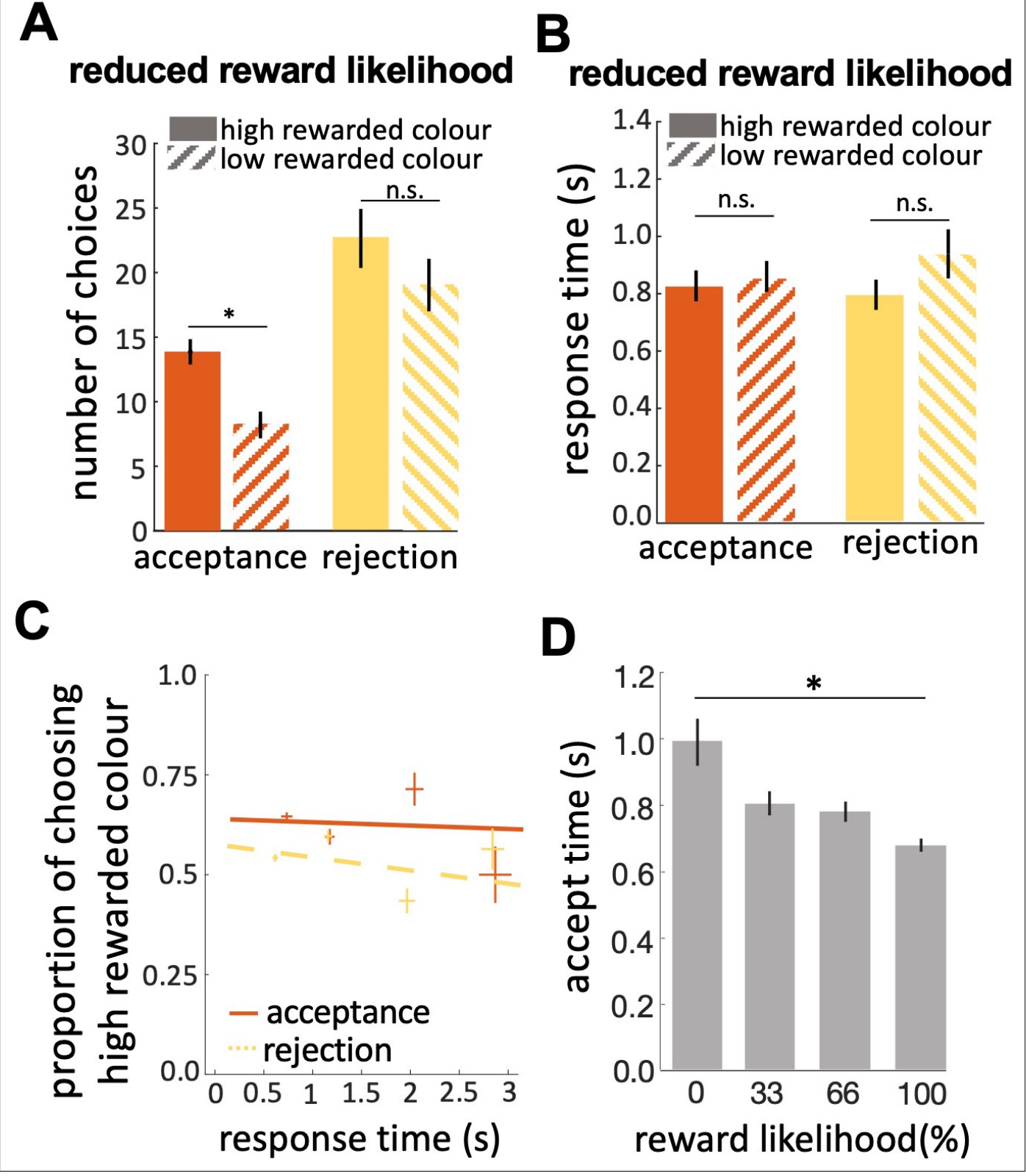

**Figure 5.** Bees' performance in the reduced reward likelihood test. (**A**) In the reduced reward likelihood test bees made more rejection than acceptance responses. Bees accepted the highly-rewarded colour more than the low-rewarded colour, but there was no difference in rejections of the two colours. (**B**) Response times did not differ for either colour or response. (**C**) CAF curves for acceptance and rejection response. The accuracy of acceptance or rejection responses did not change with response time in the reduced reward likelihood test (see *Figure 4F&G*). (**D**) Comparing acceptance times in the

*Figure 5 continued on next page*

*Figure 5 continued*

easy discrimination and reduced evidence tests allowed us to compare acceptance times for stimuli with different likelihoods of reward in training. Bees accepted the stimuli with higher reward likelihood faster. n=20, *p<0.05 and n.s., p>0.05.

reward test (*Figure 5C*). Acceptance time decreased with increasing reward expectation (*Figure 5D*; Spearman correlation, rho = 0.04, n=20, p=0.78 for acceptances; rho = –0.11, n=20, p=0.39 for rejections). Generally, our results show that bees were more likely to reject when either the available evidence or the reward likelihood was reduced.

## A minimal model for honey bee decision-making capacity

We assessed various computational sequential sampling models to explore what kinds of computation are necessary for these capacities of decision-making. We used well-established abstract models of decision-making (*Bogacz et al., 2006*). Our first model had separate accumulators for acceptance or rejection responses ($P_a, P_r$). Both accumulators receive sensory input and they provide inputs to acceptance ($A$) and rejection ($R$) command cells, respectively (*Figure 6A*). A decision is made either when one of the command cells reaches a predetermined threshold, or when a maximal decision time is exceeded. In this case, the command cell ($A \vee R$) with the highest activity determines the decision (see Materials and Methods section). It is more common in sequential sampling models to assume accumulators for specific stimuli, with each stimulus channel activating a different specific response. This structure is not biologically feasible as it would demand separate accumulators for every possible visible stimulus. Hence, we modelled accumulators for response (accept and reject) and provided both with sensory input. Simulations showed this model could neither correctly accept nor reject stimuli at above chance levels.

We then added to the model cross-inhibitory feedback signals from command cells back to the accumulators, which are constantly active during accumulating evidence at each accumulator (*Figure 6B*). In this model, as evidence accumulates in one command cell, it dampens the accumulation of evidence in the other accumulator. To build a model with a higher threshold for acceptance than the rejection response we set a stronger inhibitory connection between the reject command cell and the accept accumulator ($v_r > v_a$). This difference between the strength of cross-inhibitory feedback signals makes the model more likely to reject a stimulus whenever the evidence is insufficient. This model did indeed reject stimuli more often than accept (*Figure 6B*), but it still made an equal number of correct and incorrect choices and therefore could not discriminate between correct and incorrect decisions (*Figure 6B*).

To improve the accuracy of the model in acceptance responses we added learning cells and ($L1$ *and* $L2$) to the model (*Figure 6C*) that receive input from the sensory cells on the identity of the colours and send different inhibitory outputs to the accumulator cells (*Figure 6C*). Following a model approach by *MaBouDi et al., 2020b* $L1$ is activated when the low rewarded colours were presented to the model. $L2$ is activated by the high rewarded colour. The two accumulators receive different levels of inhibition from the learning cells based on the reward likelihood of the presented colour. If a highly-rewarded colour is presented to the model, $L2$ is activated and inhibits the reject accumulator more than the accept accumulator. This lowers evidence accumulation in the rejection accumulator. Conversely, a low rewarded colour activates $L1$ which inhibits the accept accumulator. The model with learning cells could discriminate between the high-rewarded and low-rewarded colours but in simulations, it made equal numbers of correct acceptance and correct rejection responses (*Figure 6C*). This differed from the behaviour of bees (*Figure 4B*). In summary, none of the classical sequential sampling models in *Figure 6* were able to reproduce the experimental data.

Our final model included parallel accumulators for accept and reject, learning cells and the cross-inhibitory feedback signals from the command cells (*Figure 7A*). This model could reproduce the features of bee choice behaviour (*Figures 4 and 5*): (1) In this model there was a higher threshold for acceptance than rejection, and acceptance was more accurate than rejection (*Figure 7C*); (2) When the available evidence was reduced, the model showed reduced discriminability (*Figure 7D*); (3) The model was sensitive to reward likelihood (*Figure 7E*); (4) Finally, changing evidence and reward likelihood influenced acceptance and rejection response times. By comparing the model outputs with observed bee behaviours, it becomes evident that our final model can appropriately capture the dynamic features of bee decision-making. Comparing the outputs of the different models indicates

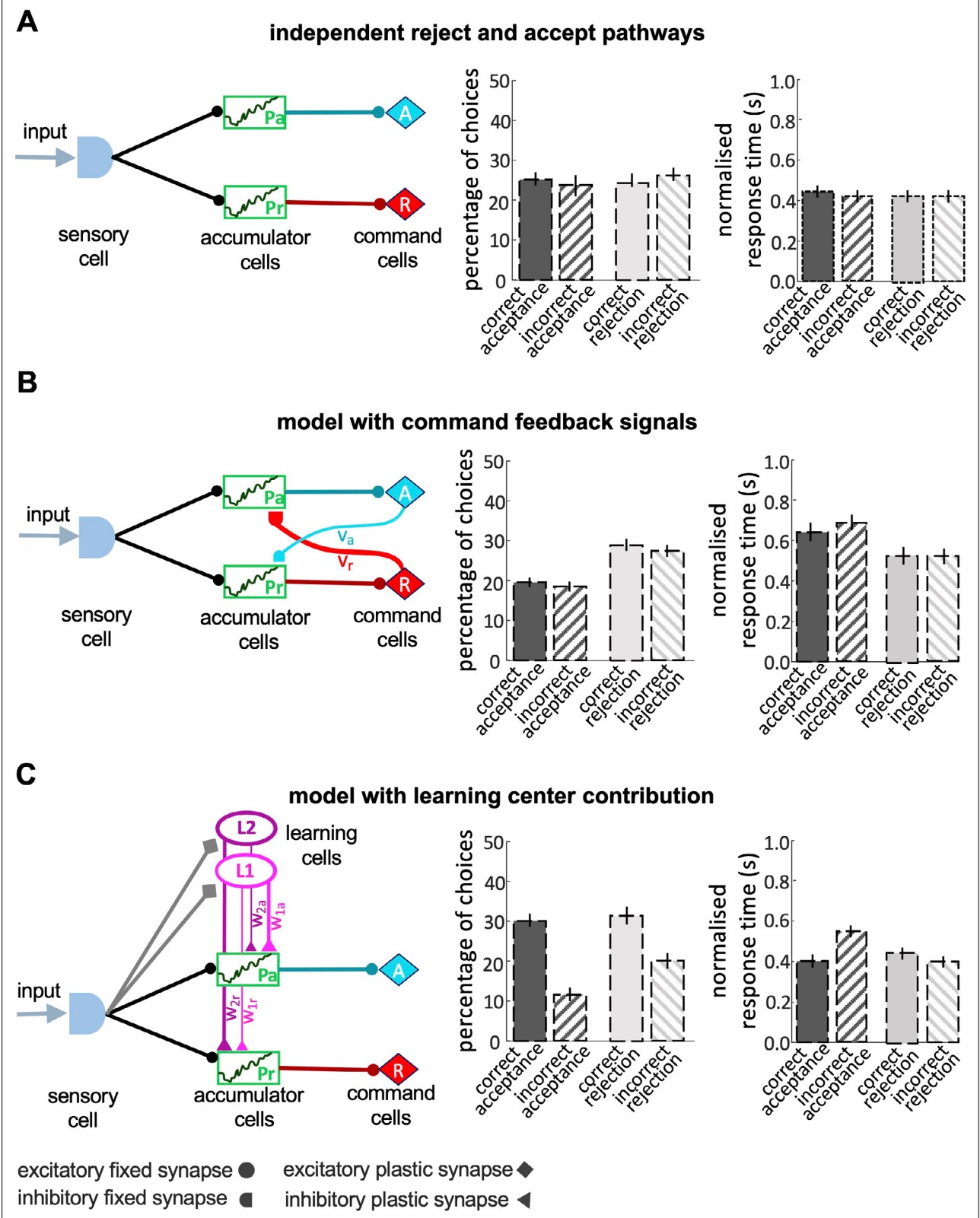

**Figure 6.** Models of decision-making. (**A**) A simple model with independent accumulators and command cells for acceptance and rejection was not able to reproduce the features of bee decisions. Correct and incorrect choices were made at equal frequency. (**B**) When cross-inhibitory feedback from the command cells was added to the model, the model was still not able to discriminate between the correct and incorrect choices, despite the number of rejections now being higher than acceptances. (**C**) A model with parallel pathways and learning cells that inhibit the accumulators with different values

*Figure 6 continued on next page*

*Figure 6 continued*

(i.e. $w_{2r} > w_{1r} \land w_{2a} < w_{1a}$) had the ability to discriminate between stimuli, but the proportion of accepting correct colours and rejecting the incorrect colours are equal.

that a parallel pathway for accept and reject accumulators is crucial in modelling bee decision-making, where both accumulators' evidence is subject to modification through learning and feedback from command cells (see decision letter section).

## Discussion

Our study has shown both sophistication and subtlety in honey bee decision-making. Honey bee choice behaviour is sensitive to the quality of the available evidence and the certainty of the outcome (**Figures 3 and 4**). Acceptance and rejection behaviours each had different relationships with reward quality and the likelihood of reward or punishment as an outcome (**Figure 5**). Acceptance had a higher evidence threshold than rejection, and the response time to accept was longer than the time to reject (**Figures 3 and 4**). As a consequence, acceptance was more accurate. We observed a large number of erroneous rejections but far fewer erroneous acceptances (**Figure 4**). Acceptance behaviour was more sensitive to reductions in reward quality and reductions in the certainty of a rewarding outcome than rejection. Correct acceptance responses were faster than incorrect acceptances (**Figure 4**) which seems counter to the well-known psychophysical speed/accuracy trade-off (**Chittka et al., 2003**; **Hanks et al., 2014**; **Heitz, 2014**). The complexity of honey bee decision-making only became apparent because we scored both acceptance and rejection behaviour. Signal detection theory has always highlighted the importance of considering both acceptance and rejection responses to understand choices but typically in animal behaviour studies rejection behaviour is usually ignored (**Figure 3**; **Trimmer et al., 2017**; **Wickens, 2001**).

How animal decision-making is influenced by sampling time has been studied in species from insects to humans (**Chittka and Niven, 2009**; **O'Connell and Hofmann, 2012**; **O'Connell et al., 2018**). The sophistication of honey bee decision-making has features in common with primates. For example, for honey bees correct acceptance decisions were faster than incorrect acceptance decisions (**Figure 4**). A similar phenomenon has been reported for primates **Churchland et al., 2008**; **Hanks et al., 2014**; **Murphy et al., 2016**; **Thura and Cisek, 2016** found that for humans in a situation requiring an urgent decision, decision accuracy decreased with increasing response times.

Primates and honey bees then appear to be behaving opposite to the expectation of the well-known speed-accuracy trade-off which predicts greater accuracy for slower decisions (**Chittka et al., 2003**; **Heitz and Schall, 2012**; **Marshall et al., 2006**; **Wickelgren, 1977**). How can this be? The speed-accuracy trade-off is considered a general psychophysical property of decision-making. It is assumed that if a signal is noisy (for any reason) evidence of the identity of the signal will build up with time. As a consequence of this decision, accuracy should increase with increasing sampling time (**Chittka et al., 2009**; **Heitz, 2014**). This psychophysical approach to animal decision-making assumes that the threshold of evidence for making a decision is fixed and does not change with the amount of time spent sampling. Ecologically that is rarely the case because sampling time incurs costs; be they energetic costs of sampling, risk of predation or opportunity costs (**McNamara and Houston, 1985**; **McNamara and Trimmer, 2019**; **Mobbs et al., 2018**). If sampling is costly and the consequences of an error are severe then a better strategy is to vary the evidence threshold for making a decision with sampling time (**Drugowitsch et al., 2012**; **Frazier and Yu, 2007**; **Malhotra et al., 2018**; **Thura et al., 2012**). One strategy under these conditions is to restrict sampling time, only to accept options for which there is very high confidence in a short sampling interval, and to reject everything else (**Chittka and Osorio, 2007**; **Fawcett et al., 2014**; **Ings and Chittka, 2008**; **Mobbs et al., 2018**; **Murphy et al., 2016**; **Trimmer et al., 2008**). A consequence of this strategy is that a very high proportion of acceptances made quickly will be correct (because the evidence threshold is high for rapid acceptance). For slower acceptances, the proportion of correct choices will be lower because the evidence threshold is lower for slower decisions. This gives an appearance of a reversed speed/accuracy relationship, but it is a consequence of the dynamic variation of the evidence threshold with increasing sampling time. The strategy of asymmetric errors that bees have taken in their decision is also predictable from the well-known optimal weighting rule from decision theory (**Freund and Schapire, 1997**; **Grofman**

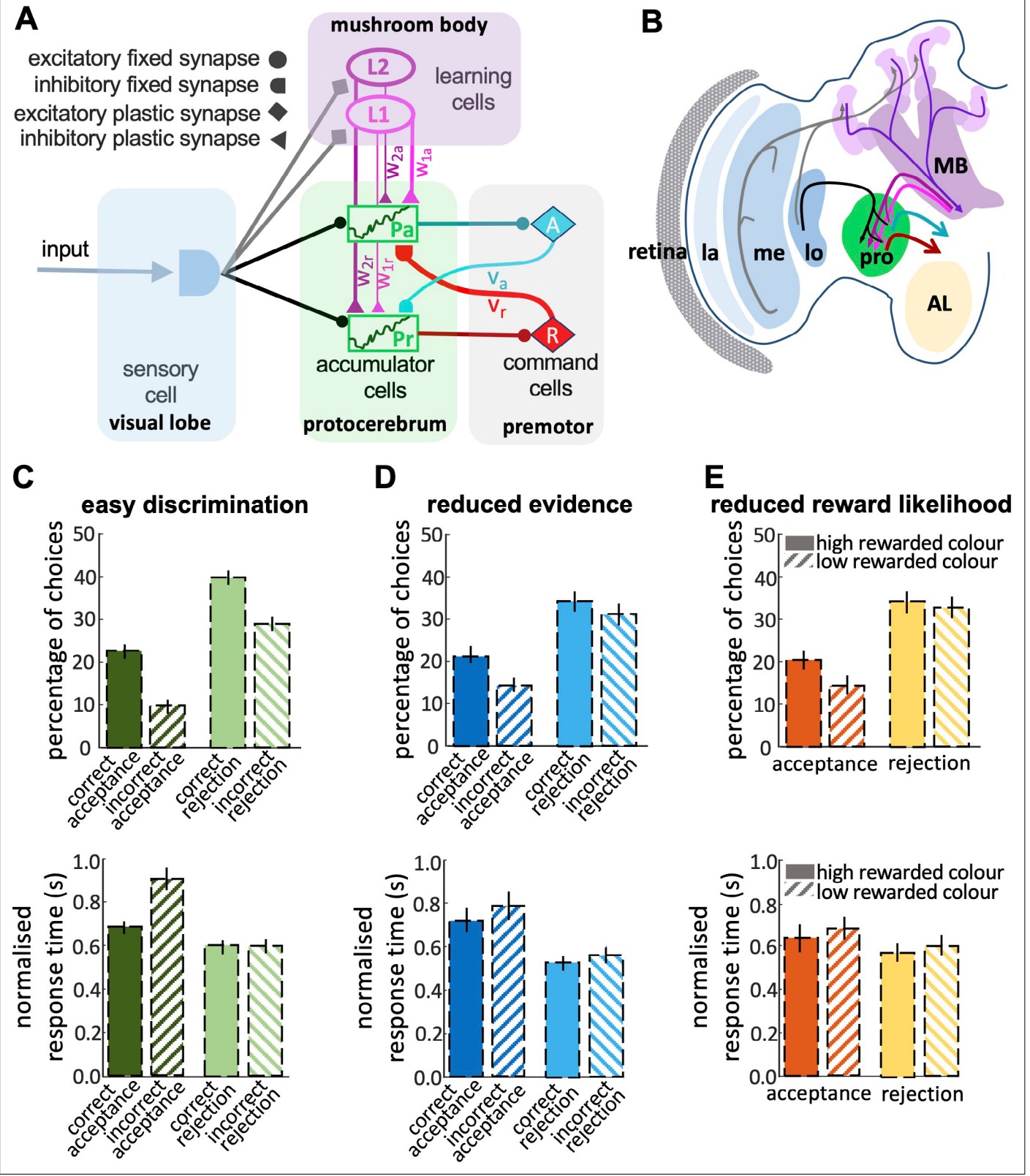

**Figure 7.** Neurobiologically-plausible model for honey bee foraging choices. (**A**) The model shows the connectivity of the components of the minimum circuitry of bee decision-making, including sensory cells, two parallel accumulators, learning cells and motor commands (See Materials and Methods). Synaptic connection classes are represented at the left-hand side. (**B**) The diagram shows a part of the insect brain involved in the decision-making process. The photoreceptors provide input from the eye to the lamina, which then sends its projections to the medulla. The medulla connects to the

*Figure 7 continued on next page*

*Figure 7 continued*

protocerebrum and, in parallel, to a third-order visual processing centre, the lobula, which then sends inputs via several tracts into the protocerebrum. In parallel, neurons in the optic lobe region (medulla and lobula) branch in the mushroom body. The anterior portions of the protocerebrum receive outputs from mushroom body output neurons (MBONs), supporting learning and memory. The output from the protocerebrum are premotor neurons. MB: mushroom bodies; AL: antennal lobe; la & me: lamina and medulla neuropils; lo: lobula; pro: protocerebrum. Our model reproduces the bees' responses to easy discrimination (**C**), reduced evidence (**D**), and reduced reward likelihood tests (**E**). The average percentage of correct choices (acceptance or rejection) made by the model bees within blocks of 25 trials. All non-overlapping SEM error bars are significantly different (p<0.05).

*et al., 1983*), the drift-diffusion model (*Marshall et al., 2017*; *Ratcliff, 1978*) and reported neural data (*Kiani and Shadlen, 2009*; *Shadlen and Kiani, 2013*). With this strategy, the number of rejections should be high overall, the number of erroneous rejections should be high and rejection accuracy less time dependent. These were all features we observed in honey bee decision-making. Hence, we propose in this study bees were following a similar time-dependent decision-making strategy (*Kiani and Shadlen, 2009*; *Malhotra et al., 2018*; *Marshall et al., 2017*; *Murphy et al., 2016*; *O'Connell et al., 2018*).

*Chittka et al., 2003* reported that for bumblebees, accuracy was positively correlated with choice time (*Chittka et al., 2003*). In this study choice time was the flight time between flowers, and they did not report an actual response time of each decision. Rejections were not reported at all. In our study recording times for all responses (acceptance and rejection) gave a more nuanced interpretation of the honey bee decision-making strategy. This emphasises the importance of recording rejections as well as acceptances.

Acceptance and rejection are fundamental aspects of animal decision-making. While rejection can be complementary to acceptance when an animal has to choose between two simultaneous choices, typically acceptance and rejection are distinct types of choices in nature. For instance, bees scan each flower independently and decide whether to land on it or reject it based on the evidence sampled from the flower, prior knowledge and other factors. Our results emphasize that acceptance and rejection are distinct features of bees' decision-making. Because rejection behaviour has a lower evidence threshold for the response it operates rather like a 'default' response to a stimulus and acceptance of a stimulus is more considered. This could be considered adaptive since accepting a flower is more risky for a bee than rejecting a flower. Rejection is performed in flight and honey bees in flight have high manoeuvrability and are only exposed to aerial predators. Accepting and landing exposes bees to far greater predation risk. Many bee predators, particularly mantids and spiders, have evolved as flower mimics and/or hide in vegetation close to flowers (*Nieh, 1993*; *O'Hanlon et al., 2014*). A foraging bee feeding on a flower is therefore exposed to greater risks than a bee in flight. Ecologically accept and reject behaviours carry different costs and benefits, and it is beneficial for bees to have separate evidence thresholds and sensitivities to evidence for acceptance and rejection.

The properties of acceptance behaviour were not fixed and were sensitive to the history of reinforcement experienced at a stimulus. Previously we have shown that in response to variable rewards bees match their choice behaviour to the probability a stimulus offers a reward (*MaBouDi et al., 2020b*). Such a probability matching strategy is the most likely ecologically rational strategy, and the best option in circumstances where the rewards offered by different options are unknown and liable to change. Here, we showed that even individual choices were influenced by the history of reinforcement (*MaBouDi et al., 2020b*). Faced with stimuli that offered both reward and punishment in training, bees' acceptance time increased, indicating the threshold for acceptance increased when there was a chance of a negative outcome from the stimulus. This shows that bees adjust how they respond to specific stimuli according to the totality of their prior experience with that stimulus.

## A neurobiological model for honey bee decision-making

Our exploration of race and LCA modelling (*Figures 6 and 7A*) showed that the simplest forms of the race model were not sufficient to capture the dynamic features of bee decision-making. Modelling all the properties of bee decisions required two channels for processing stimulus information, one of which was modifiable by learning (*Figure 7A*). These channels interacted with populations of neurons that accumulated evidence for different available options, with feedback from the command cells into the accumulator populations. Our identified model was the simplest found capable of reproducing all the qualitative features of bee decision-making (*Figure 7C, D and E*). There was a striking similarity

between the features of this minimal model and our understanding of the sensory-motor transformation in the insect brain (*Figure 7*).

In the bee brain, visual input is processed by the lamina and medulla in the optic lobes (*Figure 7B*). The medulla projects to the protocerebrum directly, and also indirectly via a third-order visual processing centre, the lobula (*Hertel and Maronde, 1987*; *Paulk et al., 2009*; *Strausfeld, 1976*; *Strausfeld and Okamura, 2007*). In parallel, the medulla and lobula project to the mushroom bodies (*Strausfeld, 1976*). The mushroom bodies are considered the cognitive centres of the brain. They receive multimodal input and support learning and classification (*Bräcker et al., 2013*; *Giurfa and Sandoz, 2012*; *Heisenberg, 2003*; *Li et al., 2017*). The protocerebrum is a complex region that is not completely characterised in honey bees, but in *Drosophila* the protocerebrum is thought to establish the valence of stimuli, whether attractive or repellent (*Das Chakraborty and Sachse, 2021*; *MaBouDi et al., 2017*; *Parnas et al., 2013*). The protocerebral regions have been hypothesised to contain 'action channels' that help to organise different kinds of behavioural output (*Galizia, 2014*). We believe the protocerebrum could feasibly contain neural populations acting like accumulators for accept or reject responses (*Aso et al., 2014*; *Dolan et al., 2019*).

That valence can be modified by learning via the outputs of the mushroom body (*Dolan et al., 2019*; *Eschbach et al., 2020*; *Lewis et al., 2015*; *Sayin et al., 2019*). These are inhibitory projections to the protocerebrum (*Mauelshagen, 1993*; *Rybak and Menzel, 1993*; *Strausfeld, 2002*). Finally, protocerebrum interneurons connect with premotor regions such as the lateral accessory lobes and central complex which generate output commands for turning and hence have the capacity to transform an accept or reject signal into an approach or avoid manoeuvre (*Cheong et al., 2020*; *Guo and Ritzmann, 2013*; *Namiki et al., 2018*; *Steinbeck et al., 2020*; *Varela et al., 2019*).

From these features of the insect brain, we can identify the functional elements needed for our minimal decision model and propose how sophisticated decisions might be possible in the insect brain (*Figure 7B*). Recent evidence from *Drosophila* has highlighted the role of the fly mushroom body in decision-making (*Groschner et al., 2018*). In a simple binary choice task, the fly mushroom body accumulated evidence on different available options using separate pools of Kenyon cells that were connected to each other by reciprocal inhibition. These experimental findings lend support to how we have mapped our model against the insect brain, but our results suggest that the fly story may be incomplete. The fly experiments did not score rejection responses, nor did they explore if the properties of the decision were sensitive to evidence quality or reward likelihood, hence the bioassay might not have exposed all the decision-making capabilities of the insect. For bees at least the mushroom body pathway cannot be the only system contributing to the decision, as dual interacting pathways were necessary (*Barron et al., 2015*; *Cheong et al., 2020*). Further electrophysiological or neurogenetic work is needed to test whether our dual pathway model is an appropriate abstraction of the insect decision system. Our model proposes a simple decision architecture that is capable of responding adaptively to the kinds of variable evidence and circumstances encountered in real-world situations. This type of model could prove of value in autonomous robotics applications (*de Croon et al., 2022*; *Kelly and Barron, 2022*; *Stankiewicz and Webb, 2021*; *Webb, 2020*).

Our study unveils the remarkable sophistication and subtlety of honey bee decision-making while emphasizing the significance of considering both acceptance and rejection responses in animal behaviour research, an aspect often overlooked in such studies. We provide compelling evidence that honey bee decision-making is influenced by the quality of available evidence and the probability of receiving a reward as an outcome. Notably, acceptance and rejection behaviours exhibit distinct characteristics, with acceptance displaying higher accuracy albeit with greater risk. Interestingly, correct acceptances were found to be faster than incorrect acceptances, contrary to the commonly observed speed/accuracy trade-off in psychophysics. Furthermore, our study, for the first time, introduces a novel and straightforward model that elucidates parallel pathways in decision-making in honey bees. This model aligns with known pathways in the insect brain and holds neurobiological plausibility. By shedding light on the neural mechanisms underlying decision-making, our findings not only provide valuable insights into honey bee behaviour but also propose a potential framework for the development of robust autonomous decision-making systems with applications in the field of robotics.

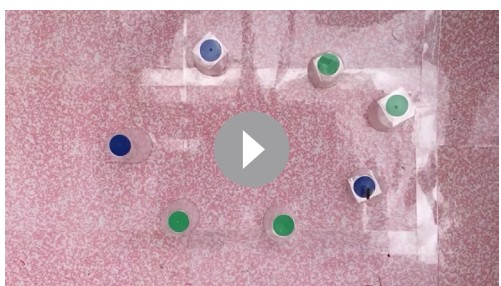

**Video 1.** Sample video of honeybee in the test. The video was captured from an overhead perspective, providing a clear view of the bees' movements within the flight arena, showcasing their reactions to various stimuli. The black lines depict the orientation of the bee's body at each frame of the video, offering further observations of their positioning and behaviour during the experiment.

https://elifesciences.org/articles/86176/figures#video1

# Materials and methods
## Bees and flight arena
Experiments were conducted at the Sheffield University Research Apiary with four standard commercial hives of honey bees (*Apis mellifera*). To source honey bees for our experiments, we provide them with a feeder containing 20% sucrose solution (w/w). Some bees visiting the feeder were given individually distinctive marks with coloured paints on their abdomen and/or thorax using coloured Posca marking pens (Uni-Ball, Japan). Experiments were performed in a (100x80 x 80 cm) flight arena made from expanded PVC foam boards with a roof of UV-transparent Plexiglas. To create a natural foraging environment for the bees, we set up the flight arena 5 m away from the gravity feeder and an additional 15 meters away from the hives. A transparent Perspex corridor (20x4 x 4 cm) provided access to the flight arena for bees. The interior walls and floor of the arena were covered with a pink random dot pattern, which created a contrast between the bees' colour and the background. This pattern was specifically designed to aid video analysis in tracking bees (*Figure 1A*, *Video 1*). All bees visiting the arena were forager bees motivated to gather sucrose for their colony. In this way, the behavioural state of bees participating in the study was standardised. The flight arena was not connected to the hives, rather for each trial bees visited the flight arena under their own volition when motivated to perform a foraging flight. Forager bees forage for their colony not for themselves and they feed in the colony prior to beginning a foraging flight. Thus, bees visiting the flight arena should have been in similar physiological and motivational states. The typical inter-visit interval by a bee was 5–10 min.

## Training and testing stimuli
Bees were trained to visit coloured stimuli inside the arena. Stimuli were disks (2.5 cm in diameter) of coloured paper covered with transparent laminate (*Figure 1A*) placed on small inverted transparent plastic cups (5 cm in height). Two additional colours intermediate between green and blue were designed for the reduced evidence test (*Figure 1B*). All colours were distinguishable for bees (*MaBouDi et al., 2020b*).

## Training protocol
During the pre-training phase, marked bees were attracted to the entrance of the arena from the gravity feeder using a cotton bud soaked in a 50% sucrose solution (w/w). Once at the gravity feeder, the bees were given more 50% sucrose solution and were gently moved to the entrance of the corridor. This process was repeated until the bee was able to fly independently to the entrance of the corridor. The bees were then trained to fly into the arena via the entrance corridor to locate drops of 50% sucrose placed on transparent disks of laminate top plastic cups. The roof of the arena was lifted to release the bees from the arena every time they were satiated. Only those bees that flew independently into the arena to feed were selected for the training phase.

Each bee was separately trained with five different coloured stimuli in a colour discrimination task for 18 bouts of training. In each trial a bee was presented with a pair of colours selected from the training stimuli (*Figure 1B*); one rewarded colour and the other punished (*Figure 1A*). During each trial, the bees were presented with four stimuli of each colour of the pair and given multiple opportunities to choose each colour until they reached satiation. Stimuli were placed randomly in the arena. Based on the reward or punishment assigned to each colour listed in the Protocol 1 and 2 during 18 trials (*Table 1*), the five different colours were each assigned a different likelihood of reward during the training trials: 100%, %66, %50, %33, and %0 of training trials (*Figure 1*). Colour pairs were organised such that in every trial one colour was rewarded and one punished (*Table 1*). For example, bees were

rewarded with stimulus S%66 in four trials (#4, #8, #13, #17 in the protocol P1; #4, #9, #12, #15 in protocol P2) and punished in two trials (#1, #10 in protocol P1; #2, #17 in the protocol P2) (*Table 1*). Thus, the likelihood of receiving a reward for stimulus S%66 during the training trials was 4/6=66%. Stimuli were rewarded with 10 *μl* sucrose solution 50% (w/w) or punished with 10 *μl* of saturated quinine hemisulphate solution.

To evaluate any effect of the innate colour preference of bees on their decision, bees were randomly assigned to one of two groups: A and B. For group A, colours were ordered as: blue = 100%, yellow = 66%, pink = 50%, orange = 33%, green = 0%. For group B, the colours were ordered as: green = 100%, orange = 66%, white = 50%, yellow = 33%, blue = 0%. Specific details on the reflectance spectrum of each colour are given in *MaBouDi et al., 2020a*.

Over 18 training trials, bees experienced all combinations of the five colours twice, with the exception that bees in training never experienced %66 rewarded paired with %33 rewarded colours. This pairing was excluded from training so that in the post-training, *reduced reward likelihood test,* we could examine how trained bees evaluate a colour pair based on the reward likelihood of colours. To control the effect of the training sequence on bees' colour preferences, bees were randomly assigned to one colour group (A or B) and one of two different sequences of training bouts (protocols P1 and P2; *Table 1*). In each training bout, bees were able to freely choose and feed from rewarded stimuli. 10 μL drops of 50% sucrose solution were replaced on depleted rewarded stimuli until the bee had fed to satiation and left the arena via the roof. Between trials, all stimuli and the arena were cleaned with soap water and then 70% ethanol and water to remove any possible pheromonal cues left by the bee. They finally were air-dried before reuse.

## Testing

Each bee was given three tests. Each test was video recorded for 120 s. In all tests, all stimuli provided 10 μl water. The *easy colour discrimination test* presented bees with the colours that had been rewarded in 100% and 0% of training trials. The *reduced reward likelihood test* presented bees with 66% and 33% rewarded colours – a combination they never experienced in training. In the *reduced evidence test* bees were given two novel colours that were similar to but intermediate to the 100% and 0% rewarded colours. The sequence of the three tests was pseudo-randomised for each bee. To maintain the bees' motivation to visit the arena, one or two refreshment trials were given between tests. In a refreshment trial, the bees were allowed to feed from 10 μL sucrose drops placed on eight disks of transparent laminate positioned in the arena. As in training, stimuli and the arena were cleaned between each test.

## Automatic bee tracking algorithm

The flight arena was equipped with an iPhone 6 camera placed at the top of the arena, 1 meter distance from the floor, facing down that captured the full base of the flight arena in the field of view (*Figure 1A*). The camera was configured to record at 30 FPS (at a resolution of 1080 pixels) in the training phase, and 240 FPS at 720 pixels in the testing phase. The first 120 s of the test and the first bout of the training phase were used to analyse bees' flights. Examples of a recorded flight path are shown in *Video 1*.

A bee's flight path was determined frame by frame extracting the x, y coordinates of the bee's body and its body orientation. From each frame, the background was subtracted using the average of the previous 50 frames. By modifying MATLAB's blob detection function with a threshold set close to the size of the bee very few candidate positions for the bee were found in each frame. We associated each pixel in each frame of the video with either a bee or the background. The bee's position at each frame obtained from the algorithm became a single point in the trajectory over time. The obtained trajectory represents the position of the bee as a function of time. An elliptic filter was applied to the frame at the position of the detected bee to evaluate the bee's body orientation. The smoothing function, 'smoothdata', was used to exclude outlier locations from the trajectory.

The flight path began when the bee entered the arena. Hovering time prior to accepting or rejecting a stimulus was assessed as the total time the bee's body was within a 5 cm radius of the centre of the stimulus (*Figure 1D*). We assumed that bees did not attend to the stimuli when flying over them at high speeds (above the height of the cups) as they did when flying between the stimuli at a similar speed, opposed to when bees were approaching the stimuli at the same height as the plastic

cups. Thus, 7% of all paths with length <0.2 s close to the edge of the focal area were excluded from analyses. A bee accepted a colour when it made contact with the colour (antennae at least contacting the platform; *Figure 1C*). This translated to an automatically count bees' landings algorithm. This algorithm counts bee's landing and utilises a threshold flight speed classifier based on the k-means algorithm that was applied to flight paths that crossed over the stimuli (*MaBouDi et al., 2021*). In this dynamic threshold determination, the speed of bees within the border of the colours was clustered into two groups: acceptance (very low-speed paths) and reject (high-speed paths). The boundary between the two groups obtained by the K-means algorithm was set as a defined rule to determine whether the bee chose or did not choose the colour.

## Flight analysis and statistics

In each test, we evaluated bees' performance from their choices during their first 120 s in the arena. Choices were scored as accepting (made a contact with colour) or rejecting a stimulus (flying away without landing). If the bee accepted the colour more likely to be rewarded in training, we considered this a correct choice. If the bee rejected the colour more likely to be punished in training, we also considered this a correct choice. Hence the bees' decision was classified into four distinct responses: (1) *correct acceptance* (CA), landing on the more rewarded colour (2) *incorrect acceptance* (IA), landing on the less rewarded colour (3) *correct rejection* (CR), rejecting the less rewarded colour and (4) *incorrect rejection* (IR), rejecting the more rewarded colour. To summarise the bees' performance in the tests, the Matthew correlation coefficient (MCC) was used as follows *MaBouDi et al., 2020a*; *Matthews, 1975*:

$$MCC = \frac{n_{CA} \times n_{CR} - n_{IA} \times n_{IR}}{\sqrt{\left(n_{CA} + n_{IA}\right)\left(n_{CA} + n_{IR}\right)\left(n_{CR} + n_{IA}\right)\left(n_{CR} + n_{IR}\right)}} \tag{1}$$

where $n_{CA}$, $n_{CR}$, $n_{IA}$ and $n_{IR}$ represent the number of CA, CR, IA and IRs for a bee in a test. The MCC has a scale from –1 to +1. High positive values indicate mostly correct acceptance and rejection choices. Negative values correspond to bees making mostly incorrect choices. Zero indicates bees choose colours randomly. A Wilcoxon signed rank test was applied to the MCC values to compare bees' performance. Finally, the relationship between bees' MCC and their scanning behaviours in the tests was evaluated by the Spearman's correlation tests. All statistical tests were performed in MATLAB 2019 (MathWorks, Natick, MA, USA). Also, to ensure the validity of our conclusions, we conducted a power analysis on the bees' performance in the experimental tests, which helped us to confirm that our sample size was sufficient (*Figure 2—figure supplement 1C*). This approach allowed us to have greater confidence in the statistical significance of our findings and to draw more accurate conclusions from our data.

## Signal detection theory

Signal detection theory (*Wickens, 2001*) was used to analyse bee decisions. Signal detection theory proposes that bees evaluate a signal (stimulus with strength x) as either rewarded or punished. We assume that the probability of either accepting or rejecting a perceived signal can be described by two distributions that are normal in shape with equal variance (*Figure 3A*). We also assume a decision criterion (*d.c.*) of the perceived signal at which the response changes from accept to reject (*Figure 3A*). From the positions of the distributions and the location of the criterion, we can estimate the expected probabilities of correct acceptances (hits) correct rejections, incorrect acceptance (false negative), and incorrect rejections (false positive; *Figure 3A*). The location of *d.c.* can be influenced by training and the experience of each signal as either punished or rewarded as well as the consequences of correct and incorrect acceptance and rejection choices (*Wickens, 2001*). Discriminability (d') is the difference in signal between the maximum likelihood of acceptance and rejection responses (*Figure 3A*). If d' is low the acceptance and rejection distributions overlap. Hence more errors are made.

Discriminability ($d'$) and the decision criteria ($d.c.$) can be calculated from the empirical measurements of hit and false positive rates as follows

$$d' = Z\left(hit\,rate\right) - Z\left(false\,positive\,rate\right) \tag{2}$$

and

$$d.c. = -\left(Z\left(hitrate\right) + Z\left(falsepositiverate\right)\right)/2 \tag{3}$$

where the function $Z\left(.\right)$ is the inverse of the standard normal cumulative distribution function (CDF). The hit rate is the ratio of correct acceptance to all acceptances ($n_{CA}/\left(n_{CA} + n_{IA}\right)$) and the false positive rate is the ratio of incorrect rejections to all rejections ($n_{IR}/\left(n_{IR} + n_{CR}\right)$).

## Modelling honey bee decision-making

We started with the simple and well-defined sequential sampling model (*Bogacz et al., 2006*; *Pike, 1966*; *Vickers, 1970*) which we adjusted to provide a better fit to experimental data for both accuracy and reaction times (*Figures 4 and 5*). Our adjustments to the sequential sampling model were constrained by the types of processing considered plausible to derive both acceptance and rejection responses through two parallel pathways.

In the model, evidence favouring each alternative ($I$) accumulated in separate accept ($P_a$) or reject ($P_r$) accumulators over time (*Figure 6A*). Biologically plausible leaky accumulators (with decay rate, $k$) were used to model the decision time which represent the duration that bees spend accumulating evidence in favour of or against a stimulus. At each time step, accept and reject accumulators send signals to the accept ($A$) and rejection ($R$) command cells, respectively. The output of command cells of accept and rejection was calculated by $A = max\left(0.1, P_a\right)$ and $R = max\left(0.1, P_r\right)$ with the baseline activity at 0.1. A decision was made either when one of the command cells reached a predetermined threshold, or when a decision was forced by exceeding a maximal assessment time in which case the decision associated with the command cell with the highest activity was chosen. The accumulation of evidence in the model is governed according to the following stochastic ordinary differential equations:

$$dP_a\left(t\right) = \left(-kP_a\left(t\right) + I\right)dt + dW_1\left(t\right), \tag{4}$$

$$dP_r\left(t\right) = \left(-kP_r\left(t\right) + I\right)dt + dW_2\left(t\right) \tag{5}$$

At time zero, the evidence accumulated $P_a$ and $P_r$ are set to zero; $P_a\left(0\right) = P_r\left(0\right) = 0$. Brownian random motions $dW_a$ and $dW_r$ are added to represent noise in input and model the random walk behaviour.

To add inhibitory feedback signals from the command cells into the accumulators (*Figure 6B*), both accept and reject accumulators actively received feedback inhibitory signals from the opposite command cells while simultaneously receiving inputs from their respective accumulators as:

$$P_a\left(t\right) = P_a\left(t\right) + dP_a\left(t\right) - v_r R\left(t\right), \tag{6}$$

and

$$P_r\left(t\right) = P_r\left(t\right) + dP_r\left(t\right) - v_a A\left(t\right) \tag{7}$$

Here $v_a$ and $v_r$ are the fraction of command outputs that inhibit the alternative accumulator.

In a previous studies (*MaBouDi et al., 2020b*; *Vasas et al., 2019*), we developed a model for the five-armed bandit task, which showed that plasticity in both the input (calyx) and output (lobes) of the mushroom body can effectively learn the history of reinforcement for different colours. This implies that the mushroom body output neurons can provide distinct inhibitory signals to the accumulator cells based on the reinforcement history of each colour. In the current study, we utilized the abstract version of learning cells from our previous work, which underwent 18 training trials for the five different colours in the five-armed bandit task, identical to what the bees experienced in this study. Building upon the model proposed in *MaBouDi et al., 2020b*, we incorporated two types of learning cells ($L_1$, $L_2$) into the model and presented the modified version in *Figure 6C*. Both learning cells received the sensory input and sent different inhibitory outputs to the accumulators based on the reward likelihood of the colours. $w_{1a}, w_{1r}, w_{2a}$ and $w_{2r}$ are the value of inhibitory signals that the accept and reject accumulators received from the learning cells ($L_1$, $L_2$) such that $w_{1a} > w_{2a}$ and $w_{1r} < w_{2r}$. The model activates the first learning cell, $r_{L_1} = \alpha I$, if the high rewarded colour is presented to the model, and activates the second learning cell, $r_{L_2} = \alpha I$, if the low rewarded colour is presented to the model. $0 \leq \alpha \leq 1$ represent the rate of the learning cells activity based on the input signal ($I$). The behaviour of learning cells and the value of the alpha were assumed and inspired by

the model presented in our previous research (*MaBouDi et al., 2020b*), that demonstrated how the reinforcement neurons modulates the strengths of the synaptic connectivity in mushroom bodies in response to both reward and punishment. Synaptic weights $w_{1a}, w_{1r}, w_{2a}$, and $w_{2r}$ were updated for each presented stimulus during training such that the accumulation of evidence in the model proceed according to the following equations:

$$dP_a\left(t\right) = \left(-kP_a\left(t\right) - w_{1a}r_{L_1} - w_{2a}r_{L_2} + I\right)dt + dW_1\left(t\right), \tag{8}$$

$$dP_r\left(t\right) = \left(-kP_r\left(t\right) - w_{1r}r_{L_1} - w_{2r}r_{L_2} + I\right)dt + dW_2\left(t\right) \tag{9}$$

where $r_{L_1}$ and $r_{L_2}$ represent the activity of learning cells $L_1$, $L_2$, respectively. Our final model, (*Figure 7A*) accumulated evidence following *Equations 8 and 9*. The accumulators received cross-inhibitory signals from the command cells according to *Equations 6 and 7*.

## Model evaluation

The models are presented with 25 trials in which high-rewarded and low-rewarded stimuli were randomly presented. Each model responded after each trial by accepting or rejecting the presented stimulus. The performance of the model was evaluated by counting the number of correct and incorrect acceptances or rejections and their corresponding response times. In addition, we normalised the time response of the model to the maximum time response of all model bees, which allowed us to make meaningful comparisons between the relative time responses of different experimental conditions and the observed time responses. This approach helped us to identify significant differences in the bees' responses to different stimuli and to gain a deeper understanding of the factors that influence their behaviour. Twenty different model bees with different random factors were examined and reported in this study. The final model could be simplified to emphasise the effect of the contributions of learning and feedback from command cells. In this way, the final model (*Figure 7A*) was also examined with learning cells inactive ($\alpha = 0$) or without the contribution of command cells by synaptic weights $v_a$ and $v_r$ set to zero. We assumed the accept and reject pathways process the input interdependently (i.e. no interaction between pathways) if $\alpha = 0$, $v_a = 0$ and $v_r = 0$.

## Acknowledgements

We thank Michael Port from Sheffield Robotics for assistance in building the testing arena. We thank Amy Bullivant for her assistance in analysing the video. HM, ND and JARM were supported by the Engineering and Physical Sciences Research Council (grant no EP/P006094/1). ABB is supported by funding by a Future Fellowship from the Australian Research Council (FT140100452), a Leverhulme Visiting Fellowship from the Leverhulme Trust and the Templeton World Charity Foundation (grant no. TWCF-2020–20539).

## Additional information

### Funding

| Funder | Grant reference number | Author |
| --- | --- | --- |
| Engineering and Physical Sciences Research Council | EP/P006094/1 | HaDi MaBouDi<br>James AR Marshall<br>Neville Dearden |
| Australian Research Council | FT140100452 | Andrew B Barron |
| Leverhulme Trust | VP1-2017-026 | Andrew B Barron<br>James AR Marshall |
| Templeton World Charity Foundation | TWCF-2020-20539 | Andrew B Barron |

The funders had no role in study design, data collection and interpretation, or the decision to submit the work for publication.

## Author contributions
HaDi MaBouDi, Conceptualization, Resources, Data curation, Software, Formal analysis, Validation, Investigation, Visualization, Methodology, Writing – original draft, Project administration, Writing – review and editing; James AR Marshall, Conceptualization, Supervision, Funding acquisition, Validation, Project administration, Writing – review and editing; Neville Dearden, Investigation, Writing – review and editing; Andrew B Barron, Conceptualization, Data curation, Formal analysis, Supervision, Funding acquisition, Validation, Investigation, Methodology, Writing – original draft, Project administration, Writing – review and editing

## Author ORCIDs
HaDi MaBouDi ⓘ http://orcid.org/0000-0002-7612-6465
James AR Marshall ⓘ http://orcid.org/0000-0002-1506-167X
Andrew B Barron ⓘ http://orcid.org/0000-0002-8135-6628

## Decision letter and Author response
Decision letter https://doi.org/10.7554/eLife.86176.sa1
Author response https://doi.org/10.7554/eLife.86176.sa2

## Data availability
Collected data have been deposited in figshare via link GitHub, (copy archived at *MaBouDi, 2023*).

The following dataset was generated:

| Author(s) | Year | Dataset title | Dataset URL | Database and Identifier |
|---|---|---|---|---|
| MaBouDi H, Marshall JAR, Dearden N, Barron AB | 2022 | Data for: How foraging honeybees make decisions | https://figshare.com/s/b7b9495beda2a0bb6db0 | figshare, b7b9495beda2a0bb6db0 |

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
