## [Editor Report]

This valuable study elucidates the honeybee's behavioral strategy to associate sensory cues with rewards of different values. Based on solid experimental evidence the study demonstrates how sensory evidence and reward likelihood quantitatively affect the decision-making process and the bees' response time. The behavioral paradigm and the proposed model could provide interesting insights for scientists studying decision-making in higher animal species.

---

## [Decision Letter]

**Decision letter after peer review:**

Thank you for submitting your article "How honey bees make fast and accurate decisions" for consideration by *eLife*. Your article has been reviewed by 2 peer reviewers, and the evaluation has been overseen by a Reviewing Editor and Michael Frank as the Senior Editor. The following individuals involved in the review of your submission have agreed to reveal their identity: Olena Riabinina (Reviewer #1); Terufumi Fujiwara (Reviewer #2).

Essential revisions:

The two reviewers found your study of interest and the results novel and exciting. However, they've raised several substantial concerns regarding the clarity of presentation and description of the methodology. In addition, we would like to ask you to either justify the number of bees and the statistical methods used or, if and where appropriate, to include additional analysis and possibly more animals. Please pay specific attention to the following points:

i) Please improve the presentation of the relevance and novelty of the data in the abstract, text, and model.

ii) Please explain the methodology of behavioral testing as well as analysis and modelling better. This point is of particular importance as it was a concern to the editors when reaching their decision to proceed to peer review.

iii) Please justify the, relatively, low number of animals used per experiment and consider applying additional statistical tests.

*Reviewer #1 (Recommendations for the authors):*

1) We recommend to the authors make a clearer statement about the relevance and novelty of the model they developed.

2) The Methods need to be clearly explained, especially those concerning the implementations of the formulae and models.

3) We are not convinced that 20 bees (3 tests each) can give sufficient data to support the conclusions. Please justify your choice of numbers.

4) The manuscript is well-written and very nicely formatted, and we love the figures in the text – thank you!

*Reviewer #2 (Recommendations for the authors):*

Line 112-113: It would be nice to have a detailed explanation of how rejection response was defined. I think it is important to clarify in the Main Text how you separated "active" rejection from just "passively" passing through the flower. Relating to this, it might be better to bring Figure 4A and the corresponding explanation earlier.

Maybe relating to this, I would like to know more about the close relationship between behaviour and the model. What does the evidence accumulation look like on behaviour? Do bees go back and forth between a pair of flowers or accumulate evidence over approaching a flower?

I did not get how the model associates 5 different flour colors with different reward probabilities. How many Learning Cells does the entire model have? Only two or more?

It would be nice to have some discussion of how the reward and punishment (i.e., sugar and quinine) signals contribute to the model. The model has input signals (flower color signals) but doesn't have reinforcement signals. Would it be possible to compare the proposed model with an RL model?

---

## [Author Response]

Essential revisions:Reviewer #1 (Recommendations for the authors):1) We recommend to the authors make a clearer statement about the relevance and novelty of the model they developed.

Thank you – we have added your suggestions to the discussion as requested on lines 420-432 which now reads: “Our study reveals the sophistication and subtlety of honey bee decision-making and highlights the importance of considering both acceptance and rejection responses in animal behaviour studies, which is typically ignored in animal behaviour studies. We demonstrate that honey bee decision-making is sensitive to the quality of available evidence and the likelihood of reward as an outcome. Acceptance and rejection behaviours are distinct, with acceptance being more accurate but riskier. Surprisingly, correct acceptances were faster than incorrect acceptances, counter to the psychophysical speed/accuracy trade-off. Additionally, for the first time, we propose a simple model of parallel decision-making in honey bees that can be mapped to known pathways in the insect brain and is neurobiologically plausible. Our study provides insights into the neural mechanisms of decision-making and proposes a potential system for robust autonomous decision-making with applications in robotics.”

2) The Methods need to be clearly explained, especially those concerning the implementations of the formulae and models.

We are sorry for not being clearer on the method. The method section now is updated with more details of the model proposed to describe the features of bees decision-making.

3) We are not convinced that 20 bees (3 tests each) can give sufficient data to support the conclusions. Please justify your choice of numbers.

Thank you for raising the issue of sample size in our study. We agree that this is an important consideration in scientific research, and we appreciate your feedback.

In order to address this concern, we conducted a power analysis on the performance of the bees in the experimental tests, which helped us to confirm that our sample size of 20 bees was sufficient to support our interpretation and conclusions. We have provided the results of this analysis in Figure 2—figure supplement 1C of our paper. This is also mentioned in the paper on lines 547-551.

While there are currently no established guidelines for the care and use of bees in research, we strive to follow the 3Rs principles (Replacement, Reduction, and Refinement) in designing our experiments. Specifically, we made every effort to utilize the minimum number of bees necessary to obtain acceptable conclusions. Recent studies have suggested that bees may experience emotion-like and pain experiences (Perry et al., 2016; Gibbons and Chittka, 2022; Gibbons et al., 2022), further highlighting the importance of minimizing any potential harm or discomfort during experimental procedures.

Furthermore, in our review of the literature on bee cognition, we found that many studies have utilized a sample size of 20 model bees in their experiments, even when conducting multiple experimental tests. While we recognize that sample size may vary depending on the specific research question and experimental design, we chose to follow this common practice in order to ensure that our results are comparable to previous studies and can be generalized to the broader population of bees.

Overall, while we acknowledge that increasing the sample size could potentially strengthen the statistical power of our study, we believe that our conclusions would not be significantly impacted by doing so. We have taken care to ensure that our methodology is sound, and our results are robust, and we are confident in the validity of our findings.

4) The manuscript is well-written and very nicely formatted, and we love the figures in the text – thank you!

Thank you very much.

Reviewer #2 (Recommendations for the authors):Line 112-113: It would be nice to have a detailed explanation of how rejection response was defined. I think it is important to clarify in the Main Text how you separated "active" rejection from just "passively" passing through the flower. Relating to this, it might be better to bring Figure 4A and the corresponding explanation earlier.

Thank you for your suggestion. We’re sorry for not being clearer on this topic. We have revised the section on lines 519-531 and included more details and references on how we identified active rejections through video analysis. We hope that this clarification will improve the comprehensibility of our study.

Maybe relating to this, I would like to know more about the close relationship between behaviour and the model. What does the evidence accumulation look like on behaviour? Do bees go back and forth between a pair of flowers or accumulate evidence over approaching a flower?

We apologise for the lack of clarity about the model. we have added more detailed explanation of the proposed model in the method section.

This study aimed to investigate the decision-making process of bees by designing a multi-options task that allowed the bees to explore each flower individually, which closely mimics their natural response in nature. The duration of time that the bees spent accumulating evidence for or against a stimulus was measured as their decision time. To model this process, we used biologically plausible leaky accumulators in both accumulators P_a_ and P_r_ as the acceptance and reception accumulators (See Figure 7A). Additionally, inputs from learning cells and command cells were added to modulate the accumulation process based on the learned stimuli and the bees' decision-making strategy. However, we attempted to present an abstract model of how bees inspect flowers and make decisions to land or reject stimuli. In the method section on lines 581- 627, variable "I" represented the value of the input signal received by the accumulators, which had two versions, one directly from the sensory cells and another modified version from learning cells. The response time of the bees in choosing or rejecting stimuli was assumed to be the time until one of the command cells reached the threshold. The number of accept or reject command cells above the threshold was considered the choice number of acceptance or rejection.

I did not get how the model associates 5 different flour colors with different reward probabilities. How many Learning Cells does the entire model have? Only two or more?It would be nice to have some discussion of how the reward and punishment (i.e., sugar and quinine) signals contribute to the model. The model has input signals (flower color signals) but doesn't have reinforcement signals. Would it be possible to compare the proposed model with an RL model?

We again apologise for not being clear about the model. We have now updated the method section with more details about the model.

In our previous research (MaBouDi et al., 2020b), we developed a model to simulate bee learning in the five-armed bandit task, showing how the learning cells in mushroom bodies can be modified by reinforcement signals to provide distinct outputs based on the history of reinforcements for different colours. Additionally, we discussed the contribution of rewards and punishments in the plasticity of mushroom bodies. In the current study, our focus was on the decision-making mechanisms, so we only presented an abstract model of the learning cells based on our previous research. However, a combined model that incorporates both learning and decision-making processes would be valuable for understanding bees' behaviour in inspecting and deciding on a set of flowers with varying reinforcement values. We plan to pursue this research question in the future.